# Immunonano-Lipocarrier-Mediated Liver Sinusoidal Endothelial Cell-Specific RUNX1 Inhibition Impedes Immune Cell Infiltration and Hepatic Inflammation in Murine Model of NASH

**DOI:** 10.3390/ijms22168489

**Published:** 2021-08-06

**Authors:** Dinesh Mani Tripathi, Sumati Rohilla, Impreet Kaur, Hamda Siddiqui, Preety Rawal, Pinky Juneja, Vikash Kumar, Anupama Kumari, Vegi Ganga Modi Naidu, Seeram Ramakrishna, Subham Banerjee, Rekha Puria, Shiv K. Sarin, Savneet Kaur

**Affiliations:** 1Department of Molecular and Cellular Medicine, Institute of Liver and Biliary Sciences, New Delhi 110070, India; dineshmanitripathi@gmail.com (D.M.T.); siskin.impreet@gmail.com (I.K.); hamda101@yahoo.com (H.S.); pinkyjuneja32@gmail.com (P.J.); anupama.parasar@gmail.com (A.K.); shivsarin@gmail.com (S.K.S.); 2School of Biotechnology, Gautam Buddha University, Greater Noida 201312, India; sumatir101@gmail.com (S.R.); rawal.prets86@gmail.com (P.R.); rpuria@gmail.com (R.P.); 3Stem Cell Biology Laboratory, National Institute of Immunology, New Delhi 110067, India; vikash2@gmail.com; 4Department of Pharmaceutics, National Institute of Pharmaceutical Education and Research, Guwahati 781122, India; vgmnaidu@gmail.com (V.G.M.N.); banerjee.subham@yahoo.co.in (S.B.); 5Department of Mechanical Engineering, National University of Singapore, Singapore 117575, Singapore; seeram@nus.edu.sg; 6Department of Hepatology, Institute of Liver and Biliary Sciences, New Delhi 110070, India

**Keywords:** inflammation, liver sinusoidal endothelial cells, immunonano-lipocarriers, non-alcoholic steatohepatitis, targeted delivery

## Abstract

Background: Runt-related transcription factor (RUNX1) regulates inflammation in non-alcoholic steatohepatitis (NASH). Methods: We performed in vivo targeted silencing of the RUNX1 gene in liver sinusoidal endothelial cells (LSECs) by using vegfr3 antibody tagged immunonano-lipocarriers encapsulated RUNX1 siRNA (RUNX1 siRNA) in murine models of methionine choline deficient (MCD) diet-induced NASH. MCD mice given nanolipocarriers-encapsulated negative siRNA were vehicle, and mice with standard diet were controls. Results: Liver RUNX1 expression was increased in the LSECs of MCD mice in comparison to controls. RUNX1 protein expression was decreased by 40% in CD31-positive LSECs of RUNX1 siRNA mice in comparison to vehicle, resulting in the downregulation of adhesion molecules, ICAM1 expression, and VCAM1 expression in LSECs. There was a marked decrease in infiltrated T cells and myeloid cells along with reduced inflammatory cytokines in the liver of RUNX1 siRNA mice as compared to that observed in the vehicle. Conclusions: In vivo LSEC-specific silencing of RUNX1 using immunonano-lipocarriers encapsulated siRNA effectively reduces its expression of adhesion molecules, infiltrate on of immune cells in liver, and inflammation in NASH.

## 1. Introduction

Non-alcoholic fatty liver disease (NAFLD) encompasses of liver pathologies, ranging from benign steatosis to inflammatory non-alcoholic steatohepatitis (NASH) to liver cirrhosis and ultimately hepatocellular carcinoma (HCC). NAFLD is the most common liver disorder in Western industrialized countries, and NAFLD has a reported prevalence of 6–35% (median 25%) worldwide [1,2]. It is now the leading cause of end-stage liver disease in Asia [3,4]. In addition to lifestyle modifications, pharmacological therapies for NASH are limited and include vitamin E, lipid lowering, and glucose-controlling drugs, but none of these is a licensed therapy for NASH [5]. 

Multifactorial mechanisms contribute to the pathogenesis and progression of NAFLD, including iron overload, inflammation, dysregulated fat metabolism, oxidative stress, gut microbiota, and angiogenesis [6]. Oxidative stress and inflammation are important mechanisms in the progression of NAFLD from steatosis to NASH to cirrhosis. Many studies have reported the over-expression of pro-inflammatory cytokines and related receptors in NASH [7,8,9]. These pro-inflammatory factors are regulated by common key transcription factors that control the progression of simple steatosis to NASH. Hence, targeting these transcription factors constitutes an important therapeutic strategy in NASH. In the current study, we focused on Runt related transcription factor, (RUNX1), which is a highly conserved transcription factor that is known to modulate several biological facets including embryological development, angiogenesis, hematopoiesis, immune, and inflammatory responses [10,11,12,13]. RUNX1 regulates inflammatory responses by controlling the NF-κB signaling pathway [11]. Liver RUNX1 is known to be upregulated by TGF-β in liver diseases including NASH [14,15]. Previous studies from our lab have shown that liver RUNX1 expression is upregulated in patients with NASH and correlates significantly with their steatosis and inflammatory grades. We also showed that RUNX1 is specifically expressed in the sinusoidal endothelial cells in NASH patients [16]. In the current study, we developed a liver endothelial cell-specific RUNX1 siRNA engineered molecule using stealth immunonano-lipocarriers (NLC). The newly developed RUNX1 siRNA engineered molecule was characterized in vitro and then investigated for its therapeutic role in the mice model of NASH. 

## 2. Results

### 2.1. Characterization of Methionine Choline Deficient (MCD) Murine Model of NASH and RUNX1 Expression 

In comparison to control mice, mice fed with MCD displayed a weight reduction of about 23% (Figure 1a). The MCD mice (after 6 weeks) showed high levels of ALT and low albumin levels as compared to the controls (Table 1). There were no significant differences in the serum triglyceride and cholesterol levels between the control group and the MCD group. MCD mice showed moderate macrovesicular steatosis, hepatocyte ballooning, and lobular inflammation (Figure 1b). The inflammatory grade was significantly higher in the MCD mice as compared to that of the control mice (*p* < 0.05). However, we did not observe any significant fibrosis in the MCD mice (not shown). 

After six weeks, the expression of RUNX1 gene was upregulated more than 10-fold in the livers of MCD models in comparison to that of the controls (Figure 1c). Cell-specific mRNA expression studies in the liver shows that RUNX1 was almost negligible in the hepatocytes in both control and MCD mice. MCD animals had enhanced expression of RUNX1 in liver nonparenchymal cells (NPCs) in comparison to that observed in the control mice. Furthermore, the NPCs (LSECs and HSCs) were cultured. LSECs were stained with a characteristic CD31 marker, while HSCs were stained with alpha smooth muscle actin (α-SMA). More than 90% of the cells stained positive for their respective cell surface markers (Appendix A). RUNX1 was found to be significantly upregulated in liver sinusoidal endothelial cells (LSECs) of MCD mice as compared to control mice. However, hepatic stellate cells (HSCs) from both the study groups showed similar expression of RUNX1 (Figure 1d). Histologically, RUNX1 expression was observed in the liver sinusoids (Figure 1e). Vegfr3 is known to be a specific marker of LSECs [17]. Moreover, co-staining with Vegfr3 illustrated vegf3-positive liver sinusoidal endothelial cells (LSECs) as the major RUNX1-expressing hepatic cell type (Figure 1f).

### 2.2. Antibody Anchored RUNX1 siRNA-Engineered Stealth Immunonano-Lipocarriers

Given the fact that RUNX1 positivity was observed in the vegfr3+ LSECs, we next synthesized engineered RUNX1 siRNA stealth immunonano-lipocarriers and to target it to liver sinusoids, we tagged it with vegfr3 antibody (Figure 2a). Specific pharmacoengineering parameters of the novel vegfr3 anchored RUNX1 siRNA-engineered stealth nano-lipocarriers were kept in mind during the synthesis of antibody anchored RUNX1 siRNA-engineered stealth immunonano-lipocarriers and those parameters were mentioned in Table 2 to achieve the targeted benefits.

The particle size of the stealth immunonano-lipocarriers with RUNX1 siRNA was found to be 263.9 nm (Figure 2b) with a Polydispersity Index (PDI) value of 0.320. No major significant changes in the size and PDI value of the nanocarriers after the attachment of primary vegfr3 antibody were observed. The zeta potential value of the RUNX1 siRNA-engineered stealth immunonano-lipocarriers before and after vegfr3 incubation were −3.39 mV (negative) and +2.53 mV (positive), respectively (Figure 2c,d). This transition of shifting charge confirms primary vegfr3 antibody attachment with improved physical stability of the particles. 

### 2.3. In Vitro Uptake, Cytotoxicity, and Efficacy of RUNX1 siRNA Immunonano-Lipocarriers in Liver Cells of MCD Mice

The uptake of RUNX1 siRNA immunonano-lipocarriers in liver cells of MCD mice was investigated by means of flourescence microscopy, as these complexes were tagged by the flourescent dye coumarin-6. The ability of both hepatocytes and liver NPCs was tested as both these cells express different levels of vegfr3. Results demonstrated a high uptake of RUNX1 siRNA immunonano-lipocarriers in the NPCs as visualized by the presence of fluorescent particles inside most of the cells after 2 h of treatment, while the hepatocytes did not exhibit an efficient uptake of these complexes. Specifically, in the LSECs, about 80% uptake (percent florescence) of NLCs was observed with vegfr3 antibody as compared to those without vegfr3 antibody, which resulted in only about 10% of the florescent signals (Figure 3a). This clearly demonstrated the efficiency of vegfr3 antibody in the uptake of NLCs in the LSECs. 

After confirming the uptake in NPCs, in vitro cytotoxicity of RUNX1 siRNA immunonano-lipocarriers at different concentrations was examined on NPCs from MCD mice by the MTT cytotoxicity test. The complexes did not cause any significant toxic effects in the investigated concentration range (Figure 3b). Furthermore, to analyze the in vitro inhibition efficacy, liver NPCs from MCD mice were treated with RUNX1 siRNA immunonano-lipocarriers (1 μM) in vitro and after 48 h of treatment, and the results showed an inhibition of about 60% as compared to vehicle-treated cells (Figure 3c).

### 2.4. In Vivo Biodistribution of RUNX1 siRNA Immunonano-Lipocarriers

After in vitro characterization, RUNX1 siRNA immunonano-lipocarriers was administered via tail vein (three injections in one week) in control and MCD murine models after six weeks. A dose of 12 μg/animal (three injections of 4 µg siRNA with nano-lipocarriers) of the RUNX1 siRNA was administered in one week. There were no obvious changes in the appearance, activity, or body weight of RUNX1 siRNA-treated animals. In vivo tissue distribution of fluorescent RUNX1 siRNA immunonano-lipocarriers was evaluated in both control and MCD mice after 2 h by using spectroflourimetry and confocal microscopy. Both the groups (control-RUNX1 siRNA immunonano-lipocarriers and MCD RUNX1 siRNA immunonano-lipocarriers) showed similar patterns of fluorescent particle accumulation with maximum levels of the fluorescence in the liver (Appendix A Appendix A, Figure 4a). To study the specific localization of the RUNX1 siRNA NLC in the liver cells, we studied a co-expression of coumarin-6 labeled nanoparticles with another well-characterized marker of LSECs, vegfr2 [17]. In the liver, these fluorescent NLCs were mainly observed in the sinusoidal endothelial cells, which is suggestive of targeted localization (Figure 4b).

### 2.5. In Vivo Efficacy of RUNX1 siRNA Immunonano-Lipocarriers

Having confirmed that RUNX1 immunonano-lipocarriers treatment was safe and localized in the liver sinusoidal vessels in the MCD animals, we next studied the in vivo efficacy of these particles in the MCD models. For this, we analyzed the gene expression of RUNX1 in the MCD livers 24 h after treatment with either vehicle or RUNX1 immunonano-lipocarriers with respect to control animals (treated with vehicle). Both vehicle and RUNX1 siRNA immunonano-lipocarriers mice showed increased expression of RUNX1 gene as compared to that in the controls. However, there was about 50% downregulation of RUNX1 gene expression in the liver tissues of RUNX1 siRNA immunonano-lipocarriers treated mice as compared to that observed in the vehicle-treated MCD mice (Figure 4c). Specifically, RUNX1 gene expression was substantially downregulated in the liver NPCs in RUNX1 siRNA immunonano-lipocarriers treated mice (Figure 4d). Further, IHC sections clearly depicted downregulation of Runx1 protein in LSECs in RUNX1 siRNA immunonano-lipocarriers treated mice as compared to that observed in vehicle (Figure 4e).

### 2.6. RUNX1 siRNA Immunonano-Lipocarriers Decrease Expression of VCAM1 and ICAM1 in LSECs

Since, we had attached vegfr3 antibody to the RUNX1 siRNA immunonano-lipocarriers, which is specifically expressed in the LSECs, we studied if RUNX1 siRNA also affected RUNX1 expression specifically in the endothelial cells. Furthermore, liver NPCs were stained with endothelial specific marker, CD31 along with RUNX1. 

In comparison to vehicle-treated MCD mice, CD31^+^ liver cells showed about 40% decline in the expression of RUNX1 in RUNX1 siRNA immunonano-lipocarriers-treated mice. The percentage of CD31^-^ RUNX1^+^ cells (Q1-1) is not different between the vehicle and RUNC1 siRNA NLC mice, while the percentage of RUNX1^+^ CD31^+^ cells (Q2-1) is significantly downregulated in the siRNA-treated mice. (Figure 5a,b). Among other major populations of NPCs, we also studied αSMA^+^ hepatic stellate cells for RUNX1 expression. We did observe an increase in the number of RUNX1 and αSMA dual positive cells in MCD mice in comparison to the controls; however, there was no significant reduction of these dual positive cells in RUNX1 siRNA immunonano-lipocarriers-treated mice (Appendix A Appendix A). Once RUNX1 knock down was confirmed in CD31^+^ endothelial cells in vivo, we next analyzed the expression of adhesion molecules on CD31^+^ LSECs using flow cytometry. The results showed that CD31^+^ cells in the RUNX1 siRNA immunonano-lipocarriers-treated mice had significantly decreased mean fluorescence intensity values for VCAM1 and ICAM1 in comparison to the CD31^+^ cells in the vehicle-treated MCD mice (Figure 5c).

### 2.7. RUNX1 siRNA Immunonano-Lipocarriers Decrease Percentage of T Cells and Monocytes/Kupffer Cells in Liver

Given the changes in the expression of adhesion molecules on CD31 positive cells of RUNX1 siRNA immunonano-lipocarriers treated mice, we next analyzed percentages of leukocytes (myeloid cells, T cells) in the liver infiltrating cells by flow cytometry. MCD mice (vehicle) had substantially increased percentages of both CD45^+^CD3^+^ T cells and CD11b^+^F4/80^+^ inflammatory monocytes/macrophages as compared to the control mice. In vivo silencing of RUNX1 with siRNA immunonano-lipocarriers markedly reduced CD45^+^CD3^+^ T cells in liver-infiltrating lymphocytes (LILs) as compared to that observed in the vehicle group (Figure 6a,b). In addition, the percentage of CD11b^+^F4/80^+^ inflammatory monocytes/macrophages in the liver was also significantly lower in the siRNA immunonano-lipocarriers-treated mice as compared to that seen in the vehicle-treated mice (Figure 6c,d). 

### 2.8. RUNX1 siRNA Immunonano-Lipocarriers Reduce Liver Inflammation 

A decrease in inflammatory cells in liver was also well evident in the H&E liver sections depicting a reduced degree of portal and lobular inflammation in the RUNX1 siRNA-treated mice versus vehicle-treated animals (Figure 7a). Since CD11b^+^ cells express high levels of CCR2 and are known to be recruited to the liver via CCL2-CCR2 interaction, we analyzed the levels of CCL2 and another pro-inflammatory cytokine, TNF-α, in the liver and also in the serum. A reduction in hepatic TNF-α and CCL2 levels in the RUNX1 siRNA immunonano-lipocarriers-treated mice was observed as compared to vehicle-treated mice (Figure 7b). However, the serum levels of these pro-inflammatory cytokines were not significantly different among the two mice groups (Figure 7c). Although higher than control mice, there was also a significant reduction in liver ALT levels in RUNX1 siRNA-treated mice as compared to vehicle-treated mice (Figure 7d).

## 3. Discussion

Inflammation and infiltration of immune cells in liver are integral steps for the development of NASH from steatotic livers. In the current study, we highlight the role of LSEC-specific transcription factor, RUNX1, as a key player involved in inflammation and immune cell infiltration in NASH using a 6-week MCD diet-induced inflammatory NASH animal model. The model showed moderate macrovesicular steatosis, hepatocyte ballooning, and lobular inflammation. The levels of TNF-α and CCL2 were also higher in the liver and serum samples as compared to control, which is suggestive of an enhanced systemic inflammation in the animal models. A low-grade systemic inflammation is a characteristic feature of NASH patients [18]. A 16-week high-fat diet-fed mice model has demonstrated similar steatosis, cell injury, portal, and lobular inflammation, as our MCD model [19,20]. Except for metabolic features, both diet HFD and MCD models have been reported to display hepatic and systemic inflammation and are comparable. The MCD diet model is a widely accepted model for the study of NASH. The MCD diet has a high sucrose and moderate fat content, but it is deficient in methionine and choline, which are essential nutrients in hepatic β-oxidation and the production of very low-density lipoprotein (VLDL) [21]. Moreover, choline deficiency leads to an impaired hepatic VLDL secretion, resulting in hepatic lipid deposition, oxidative stress, and changes in cytokines and adipokines, culminating in liver injury [22]. The primary advantages of the MCD diet are that it replicates human NASH histological features, particularly inflammation, in a relatively shorter feeding time than other dietary models. Our study focused on the transcription factor, RUNX1, which was found to be elevated in inflammatory conditions in human NASH [16], and hence, we preferred to use the MCD model as a NASH inflammatory model in our study.

Increased expression of RUNX1 in the liver significantly correlated with enhanced inflammation in the MCD models. The expression of RUNX1 was augmented in the liver NPCs specifically vegfr3-positive LSECs. Several factors could have led to an increased expression of RUNX1 in the LSECs, including an increase in steatosis. In vitro studies have shown that the treatment of primary LSECs to fatty acids acid modulates many of its physiological functions such as nitric oxide production and vasodilation [23,24,25]. In addition, mediators derived from the portal vein, steatotic hepatocytes, and other neighboring cells release inflammatory mediators in NASH that can activate LSECs [26]. A previous study has demonstrated that human retinal microvascular endothelial cells have increased RUNX1 RNA and protein expression in response to high glucose [12]. 

Negligible expression of RUNX1 was seen in the hepatocytes. This is in concordance with the observation of a low RUNX1 nuclear expression in human (Hep3B) and mouse (AML12) hepatoma cell lines and a positive expression in the HSCs [14]. A recent study also highlighted the role of RUNX1 in modulating the transcriptional activation of HSCs in NAFLD [27]. We did not observe a significant change in the RUNX1 expression in the alpha-SMA positive HSCs in the MCD animals in the flow cytometry and as well as in the primary HSCs as compared to the controls. This may be attributed to the fact that in our models, there was no significant fibrosis; hence, the correlation between RUNX1 and HSC activation/fibrosis is difficult to comprehend. It will be worthwhile to target the expression of RUNX1 in HSCs in fibrotic models. In our study, we focused on the LSEC specific knock down of RUNX1 by using vegfr3 antibody tagged immunonano-lipocarriers. Nanocarriers hold advantages such as site-specific delivery, enhanced bioavailability, low immunogenicity, and degradation over traditional si-RNAs [28]. We have not followed any conjugation chemistry, which is related to the conjugation of SC with siRNA, as our intention was not to make a lipid–antibody conjugate (by any covalent bond formation). We have rather been much more focused on the preparation of simple solid lipid nanoparticles using a modified multiple emulsification technique using an aqueous template as previously reported [29]. The Vegfr3 antibody ensured that the molecule was specifically taken up by the LSECs and not other cells, as clearly observed in the in vivo localization studies. Given the highest endocytic capacity of LSECs, the maximum uptake of immunonano-lipocarriers would be expected in the LSECs. The presence of PEG^2000^ on the liposome surface also reduced the surface binding of the immunonano-lipocarriers with plasma proteins minimizing reticuloendothelial system uptake. The immunonano-lipocarriers matrix also might have allowed a sustained release of RUNX1 siRNA in the LSECs. Although less than the in vitro system, the in vivo silencing of RUNX1 led to about 40% inhibition in the protein expression of RUNX1 after the last 24 h of injection in the CD31^+^ endothelial cells. This was validated, as only CD31^+^ cells in the NASH models showed a decrease in RUNX1 expression as compared to CD31^-^ cells. A knock down of RUNX1 in LSECs significantly reduced the cell membrane expression of CD31 as well as affected the expression of adhesion molecules, ICAM1 and VCAM1 on these cells. It has been well documented in earlier studies that during NAFLD progression, LSECs acquire a pro-inflammatory phenotype and functions [30,31]. LSECs express many adhesion molecules include ICAM1, VACM1, and E- and P-selectins that bind to their respective ligands on the leukocytes. This interaction of LSECs with leukocytes results in the accumulation of myeloid cells in the liver, leading to hepatic inflammation [32]. To this effect, we observed a significant decrease in the accumulation of both myeloid cells and T cells in the mice liver after treatment with RUNX1 siRNA-immunonano-lipocarriers, which is suggestive of the key contribution of LSEC specific RUNX1 toward increased hepatic inflammation in NASH through an upregulation of adhesion molecules in NASH. Activated LSECs not only release inflammatory mediators themselves but also activate the neighboring cells [33]. In an injured liver, infiltrated monocytes, HSCs, and LSECs are a source of CCL2, and interestingly, CCL2 is also a well-known target gene of RUNX1, indicating an important contribution of RUNX1 toward hepatic inflammation by increasing CCL2 [34,35]. A decrease in the hepatic CCL2 and TNF-α levels in RUNX1 siRNA-treated animals demonstrated that the silencing of RUNX1 is an important strategy to target the release of inflammatory mediators by LSECs in NASH liver.

## 4. Materials and Methods

### 4.1. Materials

Stearyl chloride (SC), poly (ethylene glycol)-2000 (PEG-2000), Tween 80, and coumarin-6 were procured from Sigma-Aldrich Chemical, St. Louis, MO, USA. RUNX1 siRNA and negative control siRNA were purchased from Thermofisher Scientific (MA, USA); vegfr3 was purchased from elabsciences (TX, USA). Fat-free soybean phosphatidylchloline (SPC) was obtained as a gift sample from Lipoid GmbH, Ludwigshafen, Germany. All other chemicals and solvents were of analytical reagent grade and used as received. Double-distilled water was used throughout the experiments.

### 4.2. Animals 

The study protocol was approved by the Institutional Animal Ethics Committee of the Institute of Liver and Biliary Sciences (ILBS) and was conducted according to the guidelines for the Humane Care and Use of Laboratory Animals of the National Institute of Health. Specific pathogen-free male C57BL/6 mice weighing 25–30 g were procured from Hylasco Biotechnology Pvt Ltd. (Hyderabad, India). Mice consumed food and water ad libitum and were maintained on a 12 h light/dark cycle under controlled temperature and humidity in the Animal Facility, ILBS. Mice were fed either a diet deficient in methionine and choline (MCD diet) for six weeks or standard chow diet for the same time. The MCD diet composition was 40% sucrose, 10% fat, and it was deficient in methionine and choline and was purchased from ATNT Labs, India.

Mice were randomly divided into 3 experimental groups (*n* = 12/group): the control group receiving the standard chow diet for six weeks followed by treatment with negative control or scrambled siRNA encapsulated in immunonano-lipocarriers, the vehicle group that received the MCD diet for 6 weeks followed by treatment with negative control siRNA encapsulated in immunonano-lipocarriers, and the RUNX1 siRNA NLC group that received the MCD diet for 6 weeks followed by RUNX1 siRNA encapsulated within immunonano-lipocarriers. After standard or MCD diet, mice were treated for one week by administration of negative control siRNA or RUNX1 siRNA immunonano-lipocarriers via tail vein (three injections in one week). A total dose of 12 μg/animal of the RUNX1 siRNA was administered via three injections in one week. Twenty-four hours after last treatment injection, mice were weighed and sacrificed by an overdose of anesthesia. In addition, two other groups receiving the standard diet and MCD diet were given a single injection of tagged NLC RUNX1 siRNA encapsulated in immunonano-lipocarriers and were used only for biodistribution studies (*n* = 4 each). In biodistribution studies, tissues were collected after 2 h of the injection. The complete study design is given in Appendix A Appendix A.

This given dose to the animal was determined through the confirmation of encapsulation of siRNA as per the protocol reported by Cun and Raval with slight modifications [36,37]. For the %EE determination, accurately weighed (5.0 mg) lyophilized particles matrix were solubilized in 100 µL of chloroform (for SC shell) with 400 µL of aqueous TE buffer (for SPC core). To extract the RUNX1 siRNA from organic to aqueous phase, the mixture was mixed and rotated continuously for about 2 h. Then, the mixture was centrifuged at 16,000×  *g* for 25 min at 4 °C to separate an aqueous and organic phase. Furthermore, the supernatants were diluted with the TE buffer, and concentration of the siRNA was measured using excitation wavelength: 485 nm and emission wavelength: 520 nm. We found that the RUNX1 siRNA encapsulation efficiency was around 93.0% ± 6.8. Based on those %EE values, animal doses were given.

From all the study groups, the livers were collected, sectioned, and a fraction of their lobes was used for histology, fixed in 10% formaldehyde solution for 24 h, and processed into paraffin blocks for later staining. The rest of the liver was used for the isolation of liver cells or liver-infiltrating lymphocytes (LILs). Steatosis, ballooning, and inflammation in the liver were assessed. For evaluation of inflammatory grades, specimens were classified into grades 0–3 (grade 0: none; grade 1: 1–2 foci per 200× field; grade 2: 3–4 foci per 200× field; and grade 3: more than 4 foci per 200× field) [38]. 

### 4.3. Methods

#### 4.3.1. Isolation of Primary Hepatic Cells and Liver-Infiltrating Lymphocytes (LILs)

Parenchymal cells or hepatocytes and non-parenchymal cells (NPCs) from liver were isolated [39]. Briefly, mice were intraperitoneally anesthetized (Ketamine (100 mg/kg/body weight) and xylazine (5 mg/kg/body weight)) and dissected to expose the portal vein. Liver was perfused through the portal vein with Hank’s buffer containing 1% heparin, followed by Hank’s buffer containing 0.05% collagenase solution and then excised and digested in vitro with 0.01% collagenase IV at 37 °C for 10 min. Disaggregated liver tissue was filtered using a 100 μm nylon strainer and centrifuged at 50× *g* for 5 min. The pellet contained hepatocytes, while NPCs were found in the supernatant. 

For hepatocyte culture, the above pallet was washed twice with DMEM with 10% FBS and tested for its viability with trypan blue stain. Dead hepatocytes were eliminated by 100% percoll centrifugation at 50× *g* for 5 min 4 °C. More than 80% cell viability was used to culture hepatocytes. Then, 4.5 × 10^6^ cells were seeded per 6-well supplemented with DMEM-F12 with 10% FBS and 1% ITS for 6–12 h for adherence followed by DMEM-F12 with 1% ITS for maintenance.

For NPC culture, the above supernatant was centrifuged at 800× *g* 10 min and re-suspended in PBS. To further eliminate hepatocytes, again, suspension was centrifuged at 50× *g* for 5 min 4 °C, and the supernatant was centrifuged at 800× *g* 10 min 4 °C. The resultant pellet was resuspended in 10 mL PBS. Next, the LSECs were isolated by using 50%/25% percoll density gradient centrifugation at 800× *g* 25 min 4 °C with acceleration and deceleration as 0. The ring between the 50%/25% percoll contains LSECs with Kupffer cells, while the top most buffy layer contained HSCs. The LSEC ring was extracted, washed with PBS, and plated onto a 100 mm Petri dish for 20–35 min with RPMI supplemented with 10% FBS to eliminate Kupffer cells. Next, very gently, the cell soup was plated on a rat tail collagen-coated well plate for 0.45–1 h followed by replacement of media with RPMI supplemented with 10%FBS and 1% endothelial cell growth factors. The top most buffy layer containing HSC was isolated was washed twice with PBS and cultured with IMDM with 10% FBS.

For the isolation of LILs, liver was minced into 1 mm pieces and digested using collagenase IV for 30 min at 37 °C. The digested liver extracts were filtered through 70 μm cell strainers and centrifuged at 500× *g* for 5 min. The resulting cell pellet was resuspended in 10 mL of 35% Percoll containing 100 U/mL heparin and centrifuged at 700× *g* for 15 min at room temperature. The cell pellet containing the leukocytes were collected and resuspended in 3 mL of red blood cell lysis solution (155 mM NH4Cl, 10 mM KHCO3, 1 mM EDTA, 170 mM Tris, pH 7.3). After incubation for 3 min on ice, the cells were washed twice in RPMI 1640 containing 5% fetal bovine serum.

#### 4.3.2. Characterization of LSECs and HSCs

The cultured LSEC and HSCs were assessed for their purity by performing immunofluorescence staining with their cell specific surface markers. Briefly, the cells were cultured in their respective media for 24 h. Furthermore, the cells were fixed with 4% paraformaldehyde for 15 min at room temperature, washed thrice with PBS followed by overnight incubation at 4 °C with primary antibody CD31 for LSEC and α-SMA for HSCs. The next day, cells were washed thrice with PBS and incubated for 2 h with secondary antibody, which was followed by washing and adding DAPI to stain the nucleus. The antibodies are given in Appendix A Appendix A. The cells were properly washed and visualized under an inverted fluorescence microscope (Nikon Instruments Inc., Melville, NY, USA).

#### 4.3.3. Development and Characterization of RUNX1 siRNA-Engineered Stealth Immunonano-Lipocarriers

##### Preparation of RUNX1 siRNA Nano-Lipocarriers

RUNX1 siRNA-engineered stealth immunonano-lipocarriers was formulated using a cold-High Shear Homogenization (HSH) technique using digital High Shear Homogenizer. 

Given the hydrophilicity of both the moieties, RUNX1 siRNA was first encapsulated inside the core of SPC matrix (2% *w/w* in Tween 80 containing cold aqueous solution) at a weight ratio of 1:6. Then, primary emulsion was prepared by adding this RUNX1 siRNA encapsulated SPC core into the outer shell of coumarin-6 labeled SC (225 µL which is equivalent to 200 mg of SC) under magnetic stirring (RCT basic, IKA India Pvt. Ltd., Bengaluru, India) at 1500 rpm for 15 min. The SC-mediated oil phase (o) acted here not only as a continuous phase but also a former shell as well, whereas the siRNA-encapsulated SPC mediated aqueous phase (w_1_) acted here both as the dispersed phase and core-forming agent inside the SC shell matrix structure [29]. Furthermore, this cold primary emulsion was transferred into secondary cold aqueous phase containing 0.15% *w/v* PEG-2000 solution (stealthing agent) under ice-bath condition with high-speed homogenization (T-10 Digital Ultra-Turrax^®^, IKA India Pvt. Ltd., Bengaluru, India) technique at 10,000 rpm for 3–5 min in order to make RUNX1 siRNA-engineered stealth immunonano-lipocarriers dispersion. Then, the resultant dispersions were purified by ultrafiltration technique to get rid of larger size particles, lumps, or any kind of pathological contaminations [40]. The obtained nanodispersion was kept at 4 °C for further analysis.

##### Antibody Anchored RUNX1 siRNA-Engineered Stealth Immunonano-Lipocarriers 

RUNX1 siRNA-engineered stealth immunonano-lipocarriers dispersion was further incubated with vegfr3 primary antibody (50 ng/µL in Tris-EDTA buffer) for 48 h at 4 °C temperature in order to finally obtain vegfr3 anchored RUNX1 siRNA-engineered stealth immunonano-lipocarriers for targeted delivery in liver sinusoids. RUNX1 siRNA-engineered stealth immunonano-lipocarriers without vegfr3 were also used in the in vitro uptake experiments as control.

##### Particle Size, Polydispersity Index (PDI), and Zeta Potential

Before and after vegfr3 incubation, RUNX1 siRNA-engineered stealth immunonano-lipocarrier dispersion was evaluated for mean particle size (z-average), PDI, and zeta potential analysis by the dynamic laser light scattering method using a Zetasizer (Nano ZS-90, Malvern Instruments, Worcestershire, UK). The stability of the dispersion systems was monitored by zeta potential values at room temperature. 

##### Cell Viability Assays with RUNX1 siRNA-Engineered Stealth Immunonano-Lipocarriers 

NPCs from MCD mice were seeded in serum-containing media in 96-well plates at the density of 1.5 × 10^3^ cells/well at 37 °C for 24 h. RUNX1 siRNA immunonano-lipocarriers was added at concentrations of 1 μM. After incubation for 2 h, the medium was removed, and the cells with entrapped siRNA were observed using an inverted fluorescence microscope (Nikon Instruments Inc., Melville, NY, USA). For cell viability assays, cells were treated with increasing concentrations of RUNX1 siRNA immunonano-lipocarriers or 1 μM negative control siRNA encapsulated in immunonano-lipocarriers (vehicle) for 24 h. Cell viability was assessed by adding MTT (3-(4,5-dimethylthiazol-2-yl)-2,5-diphenyl tetrazolium bromide) solution in phosphate-buffered saline (PBS) to a final concentration of 0.5 mg/mL. Then, the plates were incubated at 37 °C for an additional 4 h, and the MTT–formazan crystals were solubilized in 1N isopropanol/hydrochloric acid 10% solution at 37 °C for 20 min. Absorbance was measured at 570 nm using a Bio-Rad 550 microplate reader (Bio-Rad Laboratories). Percentage cell viability was calculated as 100% × (absorbance of the treated wells—absorbance of the blank wells)/(absorbance of the control (untreated) wells—absorbance of the blank control wells). MTT experiments were performed in triplicates and repeated at least 3 times. 

#### 4.3.4. In Vivo Biodistribution and In Vitro Uptake Studies

For biodistribution, animals were euthanized 2 h after the last injection of coumarin-6 labeled RUNX1 siRNA immunonano-lipocarriers, for the collection of liver, duodenum, spleen, lung, and kidney tissues. The tissues were weighed and homogenized with physiological saline, and the fluorescence intensity was detected at 520 nm using a spectroflourimeter. Some part of the tissues was also observed under a flourescence microscope for the visual detection of labeled nanoparticles. To identify the specific localization of the nano-lipocarriers in liver in these animals, 8 μm sections from frozen MCD liver tissue blocks were cut using the cryotome. The tissue sections were fixed with a pre-cooled acetone (−20 °C) for 10 min. The slides were washed with phosphate-buffered saline (PBS) for 2 times, 5 min each. Next, the slides were incubated in 0.3% H_2_O_2_ solution in PBS at room temperature for 10 min to block endogenous peroxidase activity and were washed with PBS two times. Furthermore, the sections were blocked with 3% bovine serum albumin for 15 min in a humidified chamber at room temperature. The blocking reagent was drained off from the slides, and primary antibody was added to the sections on the slides overnight at 4 °C. The next day, the slide was rinsed with PBS two times, and secondary antibody was added to the sections for 1 h in a humidified chamber at room temperature. Next, the slides were washed and mounted with the glycerol and visualized under a confocal microscope (Nikon confocal microscope model A1R HD25).

For in vitro uptake studies, we cultured primary LSECs, and upon 80% confluency, the cells were treated with RUNX1 siRNA NLC with and without vegfr3 antibody for 2 h. The images were captured using an inverted fluorescence microscope (Nikon Instruments Inc., Melville, NY, USA). The percent fluorescent intensity was calculated after normalization of the background fluorescence noise.

#### 4.3.5. Real-Time qRT-PCR

The total RNA was extracted from whole liver tissue and cells using an RNA Sure mini kit (Nucleopore, Genetix, Delhi, India; cat. no. NP-84105). LSEC and HSCs were cultured for 24 h with suitable media (LSECs; RPMI supplemented with 10% FBS and 1% endothelial cell growth factors, HSCs; IMDM with 10% FBS) under 5% Co2 incubator. The RNA was isolated by grouping two wells of a 6-well plate; per well, 250 µL of trypsin was used to trypsinize the cells. The resultant cell pallet was lysed using the lysis buffer, followed by manufacturer’s instructions. RNA was quantified at 260/280 nm with Thermo Scientific Nanodrop 2000 Spectrophotometer. The absorption ratio A260 nm/A280 nm between 1.90 and 2 was taken into consideration for cDNA preparation. First-strand cDNA was synthesized from 1 µg of total RNA with reverse transcriptase (Thermo Scientific Verso cDNA synthesis kit) according to manufacturer instructions. Quantitative real-time PCR was performed with SYBR green PCR master mix (Fermentas Life Sciences, Bengaluru, India) on the ViiA7 instrument PCR system (Applied Biosystems, CA, USA). A dissociation curve was generated at the end of each PCR to verify that a single DNA species was amplified. The following cycling parameters were used: start at 95 °C for 5 min, denaturing at 95 °C for 30 s, annealing at 60 °C for 30 s, elongation at 72 °C for 30 s, and a final 5 min extra extension at the end of the reaction to ensure that all amplicons were completely extended and repeated for 40 amplification cycles. Mouse RUNX1 primer pairs used in the study were FP: CCTCCGGTAGTAATAAAGGCTTCTG, RP: CCGATTGAGTAAGGACCCTGAA. The mouse GAPDH gene was used as the housekeeping gene, FP: GTTGTCTCCTGCGACTTCA, RP: GGTGGTCCAGGGTTTCTTA.

#### 4.3.6. Immunohistochemistry Analysis

For immunohistochemistry (IHC), mice liver tissues were fixed and stained as per standard protocols. First, 2 μm tissue sections were fixed in 4% phosphate-buffered formaldehyde solution and embedded in paraffin. They were cut with a sliding microtome for histology. Sections were deparaffinized, rehydrated, and incubated for 7 min with trypsin at 37 °C. Thereafter, endogenous peroxidase was blocked for 20 min with hydrogen peroxidase in methanol. Subsequently, they were incubated overnight at room temperature with mouse primary antibodies. The antibodies used for IHC are given in Appendix A Appendix A. Thereafter, the specimen was incubated with the PolyExcel Target Binder for 10 min followed by a PolyExcel HRP labeled polymer using recommended 10 min incubation (PathNSitu Biotechnologies, Secunderabad, India). Staining was completed by a 5–10 min incubation with 3, 3′-diaminobenzidine (DAB) substrate–chromogen, which resulted in a brown-colored precipitate at the antigen site. Counterstaining was done using hematoxylin.

#### 4.3.7. Flow Cytometry Analysis

Flow cytometry was done for both the liver cells and LILs isolated from different study groups. For cell surface staining, the cells were isolated, and the cell suspension was incubated with fluorophore-conjugated antibodies for 3 h at 4 °C. Furthermore, the cells were washed with PBS twice and fixed with paraformaldehyde in PBS and analyzed. For intracellular staining, the cells were isolated and permeabilized with 0.1% tritonx 100 and incubated with fluorophore-conjugated antibodies for 3 h at 4 °C, washed with PBS twice and fixed with paraformaldehyde in PBS and analyzed by BD FACS Aria III (BD Biosciences) using DIVA software, and gated as described in the Supplementary Material (Appendix A Appendix A). A minimum of 1 million events was acquired. The antibodies used for flow cytometry are given in Appendix A Appendix A. 

#### 4.3.8. ELISA Assays for TNF-α and CCL2

The levels of TNF-α and CCL2 were analyzed in the liver and serum samples by ELISA using Thermofisher Scientific ELISA kits as per manufacturer’s instructions (ThermoFisher Scientific, Invitrogen, Bengaluru, India). 

#### 4.3.9. Statistical Analysis

Data are expressed as mean ± standard deviation. Statistical significance is accepted as *p* ≤ 0.05. Student’s unpaired t-tests have been used to analyze and compare the groups. 

## 5. Conclusions

To summarize, the study describes the formulation of a stable immunonano-lipocarrier encapsulated siRNA for LSEC-specific silencing of RUNX1. An increased expression of RUNX1 imparts a pro-inflammatory phenotype to LSECs during NASH progression. Treatment with the immunonano-lipocarrier encapsulated siRNA results in a decrease in RUNX1 expression in LSECs and reduces the expression of adhesion molecules, causing decreased infiltration of myeloid and T cells in liver, and finally attenuating inflammation in NASH liver. The study highlights the significance of a nanodelivery system for in vivo cell-specific gene silencing and proposes it as a novel strategy to target LSEC-associated inflammation in NASH. 

## Figures and Tables

**Figure 1 ijms-22-08489-f001:**
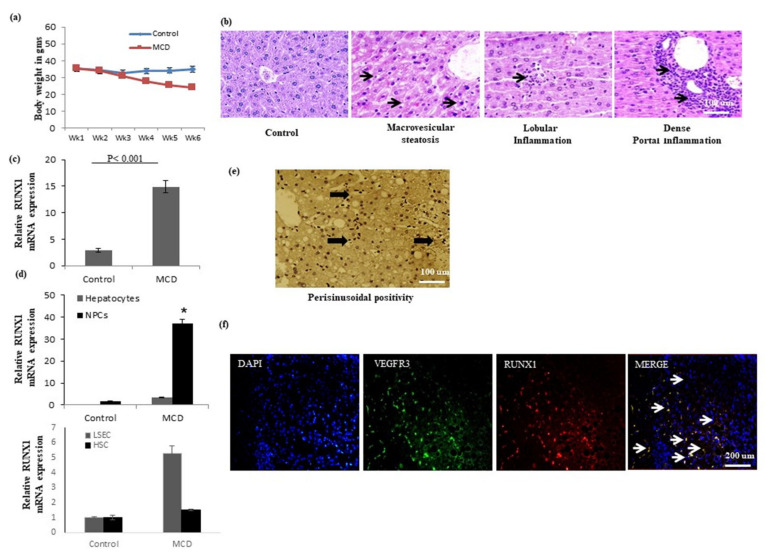
RUNX1 expression in animal models of steatosis and NASH. (**a**) Body weight (in grams) of mice fed with standard diet (control) or methionine choline diet (MCD). (**b**) Liver histology of control and MCD mice. The first image in the figure is a representative image of a control mice liver, while the other images are from MCD mice liver (arrows) showing evident steatosis, lobular, and portal inflammation. (**c**) Relative RUNX1 mRNA expression in liver tissues of control and MCD mice. (**d**) Relative RUNX1 mRNA expression in liver cells (hepatocytes, NPCs, LSECs, and HSCs) of control or MCD mice. (**e**) Immunohistochemical image (arrows) indicating RUNX1 nuclear expression in sinusoidal endothelial cells. (**f**) Co-staining of RUNX1 with Vegfr3 in MCD mice liver. Blue color indicates nucleus, green and red color indicates Vegfr3 and RUNX1 expression respectively, arrows indicating yellow color are the cells expressing both vegfr3 and RUNX1 expression. Data represented as mean ± SD, *n* = 4, * represents *p* value < 0.05 between controls and MCD.

**Figure 2 ijms-22-08489-f002:**
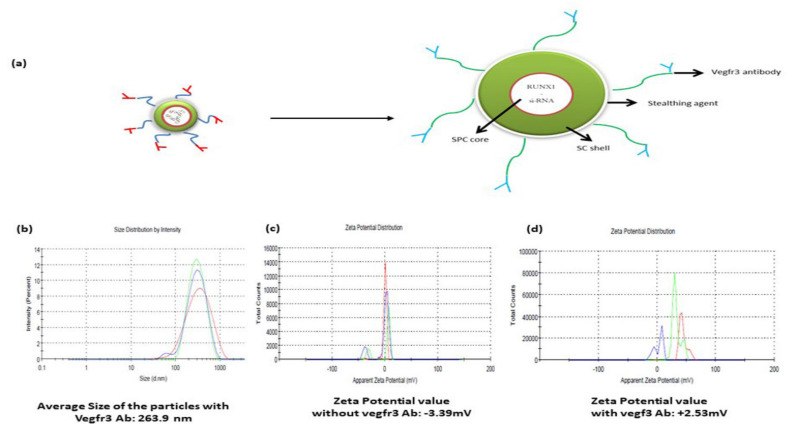
Development and characterization of RUNX1 siRNA-engineered stealth immunonano-lipocarriers. (**a**) Design sketch of RUNX1 siRNA-engineered stealth immunonano-lipocarriers for in vivo cell-specific gene silencing. (**b**) Mean particle size of pharmacoengineered RUNX1 siRNA nanolipocarriers. (**c**) Mean zeta potential value without vegfr3 and (**d**) Mean zeta potential value with vegfr3. Different peaks in the graphs are representing different replicates of samples. SC: stearyl chloride; SPC: soybean phosphatidylchloline.

**Figure 3 ijms-22-08489-f003:**
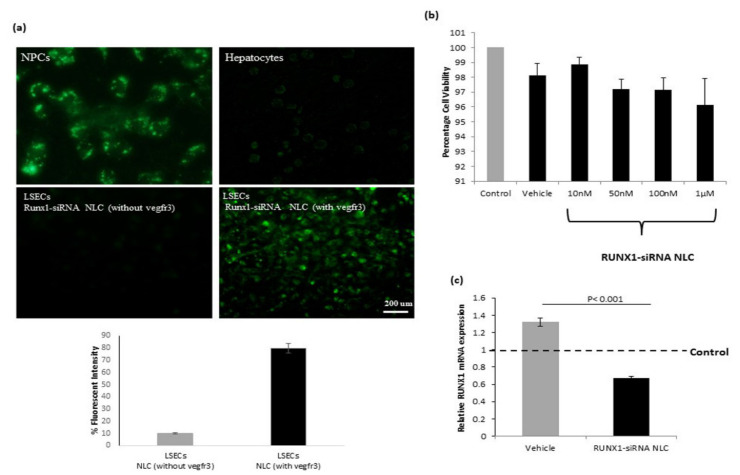
In vitro uptake, cytotoxicity, and efficacy of RUNX1 siRNA nano-lipocarriers (NLC). (**a**) Florescence microscopic images showing uptake (green color) in liver nonparenchymal cells (NPCs), hepatocytes, and LSECs from MCD mice after incubation with RUNX1 siRNA NLC (1 µM). RUNX1 siRNA NLC without vegfr3 antibody was used to show antibody-directed uptake in the LSECs. (**b**) Cytotoxicity induced by RUNX1 siRNA in liver NPCs at different concentrations as estimated by MTT assay. Percentage viability of NPCs was evaluated as 100 × (absorbance of the treated wells—absorbance of the blank wells)/(absorbance of the control wells (untreated)—absorbance of the blank wells). MTT experiments were performed in triplicates and repeated three times. Vehicle represents cells treated with negative control siRNA encapsulated in NLC. (**c**) Relative RUNX1 mRNA expression in liver NPCs from MCD mice treated with vehicle or RUNX1 siRNA NLC (1 µM) in vitro with respect to control NPCs (untreated). Data represented as mean ± SD, *n* = 4.

**Figure 4 ijms-22-08489-f004:**
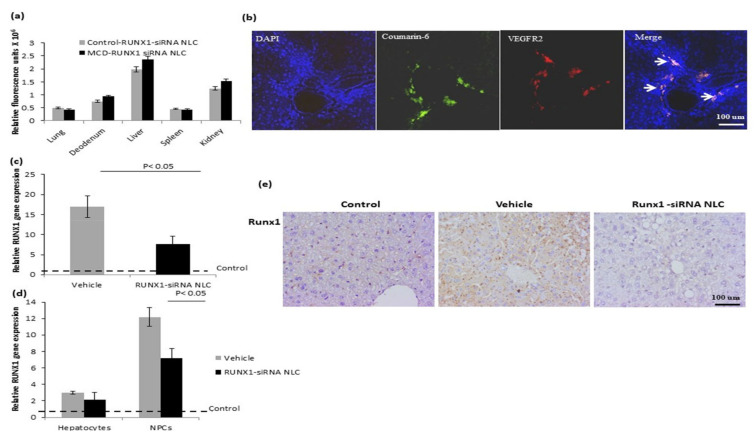
In vivo biodistribution and efficacy RUNX1 siRNA NLC. (**a**) Relative fluorescence units (flourescence intensity at 520 nm) of coumarin-6 tagged RUNX1-NLC in control and MCD mice after 2 h of tail vein delivery. Data represented as mean ± SD, *n* = 4. (**b**) Confocal florescence microscopic images showing the localization of RUNX1 siRNA NLC in the liver sinusoidal endothelial cells around the portal area stained with VEGFR2 antibody in MCD mice liver tissue sections after 2 h of tail vein delivery. Blue color indicates the nucleus, green color indicates the cells stained with coumarin-6 dye of NLC, red color indicates the cells showing VEGFR2 expression and (arrows) golden color indicates the cells showing colocalization of VEGFR2 and NLC. (**c**) Relative RUNX1 mRNA expression in liver in vehicle (MCD mice treated with NLC only) and in RUNX1 siRNA NLC-treated MCD mice (1 µM) in vivo. (**d**) Relative RUNX1 mRNA expression in liver cells in vehicle (MCD mice treated with NLC only) treated with RUNX1 siRNA NLC with respect to controls (control mice treated with NLC) in vivo. (**e**) Immunohistochemical image showing LSEC specific RUNX1 expression in liver sections of control, vehicle, and RUNX1 siRNA NLC. For gene expression studies, liver tissues were collected 24 h after the last injection of the RUNX1-NLC. Data represented as mean ± SD, *n* = 5.

**Figure 5 ijms-22-08489-f005:**
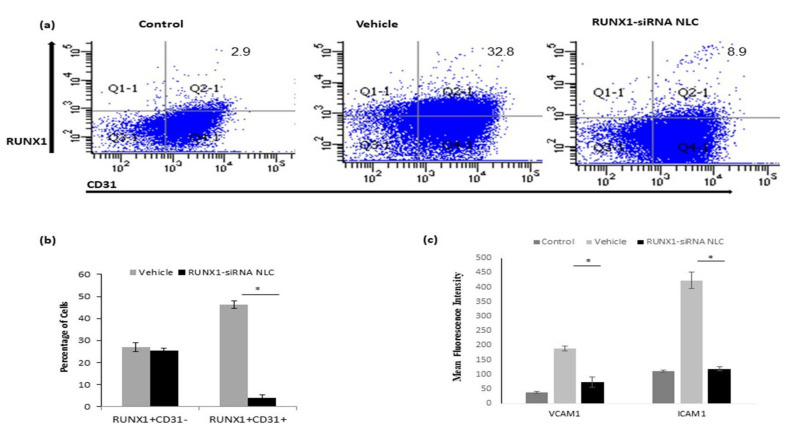
Protein expression of RUNX1 in liver endothelial cells in vivo. (**a**) Representative dot plots showing in RUNX1^+^CD31^+^ and RUNX1^+^CD31^−^ cells. (**b**) Bar diagrams depicting average percentage of RUNX1^+^CD31^+^ and RUNX1^+^CD31^−^ cells in vehicle and RUNX1 siRNA NLC mice. (**c**) Mean fluorescence intensity of VCAM1 and ICAM1 in CD31^+^ cells in vehicle and RUNX1 siRNA NLC mice with respect to controls. Data represented as mean ± SD, *n* = 5. “*” represents *p* < 0.05 versus controls.

**Figure 6 ijms-22-08489-f006:**
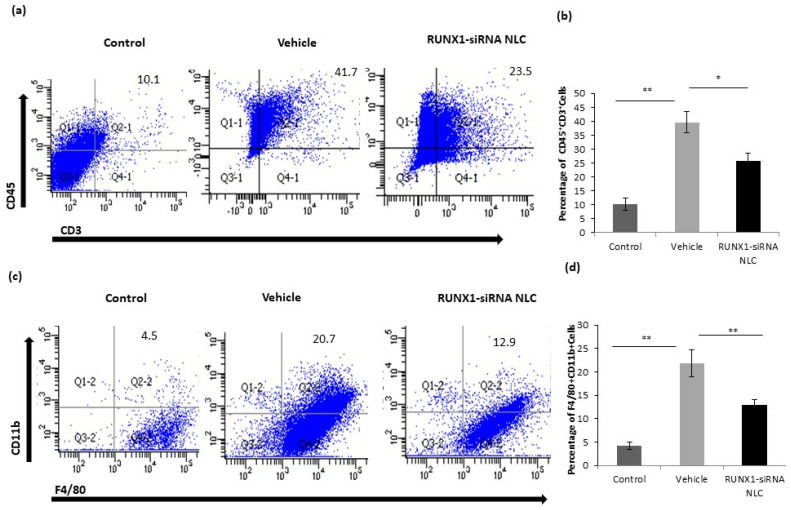
Immune modulation in RUNX siRNA NLC-treated mice. (**a**,**b**) Representative dot plots and bar diagrams showing percentages of CD45 + CD3 + leukocytes in liver-infiltrating lymphocytes in control, vehicle, and RUNX1 siRNA NLC mice. (**c**,**d**) Representative dot plots and bar diagrams percentages of CD11b + F4/80 + leukocytes in liver-infiltrating lymphocytes in control, vehicle, and RUNX1 siRNA NLC mice. Data represented as mean ± SD, *n* = 5 * *p* < 0.05, ** *p* < 0.001.

**Figure 7 ijms-22-08489-f007:**
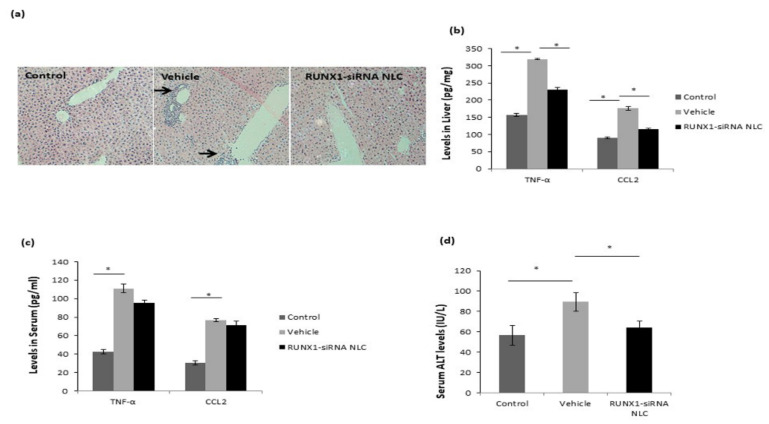
RUNX1 siRNA mice show reduced liver inflammation. (**a**) H&E (10×) and CD45 (20×) immunohistochemistry of liver sections showing inflammatory cells (arrows) in the lobular and portal areas of liver in vehicle and RUNX1 siRNA NLC mice. (**b**) Levels of TNF-α and CCl2 (pg/mg protein) in liver tissue homogenates of control, vehicle, and RUNX1 siRNA NLC mice. Standard curve was drawn using standards provided in the kit, and each analyte concentration was calculated from the standard curve. The assays were normalized to 1 mg liver protein. (**c**) Levels of TNF-α and CCl2 (pg/mL) in serum samples of control, vehicle, and RUNX1 siRNA NLC mice. (**d**) Serum ALT Levels (IU/L) in control, vehicle, and RUNX1 siRNA NLC mice. Data represented as mean ± SD, *n* = 4. * *p* < 0.05.

**Table 1 ijms-22-08489-t001:** Serum and tissue parameters in control and MCD animals.

Characteristics	Control	MCD
ALT (IU/L)	56.7 ± 9.7	89.4 ± 8.9
Albumin (g/dL)	3.6 ± 1.29	1.6 ± 0.12
Triglyceride (mmol/L)	100 ± 4.9	102 ± 3.6
Cholesterol (mg/dL)	67 ± 1.8	62 ± 2.3
Inflammatory grade	0.2 ± 0.4	2.25 ± 0.5

**Table 2 ijms-22-08489-t002:** Pharmacoengineering parameters of RUNX1 siRNA-engineered stealth immunonano-lipocarriers.

Engineering Parameters	Desired Optimum Values/Features	Achievable Benefits
Size	<300 nm	Less size with a high surface area
Shape	Spherical	Improved cellular delivery
Matrix	Lipids (SC-shell) and phospholipids (SPC core)	Sustained RUNX1 siRNA release for improved biodistribution profile of engineered matrix deliverables
Surface charge	Positive (after vegfr3 anchoring)	Improved physical and biological stability, avoid agglomeration
Recognition by macrophages	Hydrated stealth (PEGylated)	Avoid recognition by other organ macrophages
Surface ligand	Primary vegfr3 antibody	To trigger surface antigen–antibody interactions
SC-shell	Solid and hydrophobic	To easily cross-cellular membrane
SPC-core	Amphiphilic, vesicular, and soft matrix	Promote sustain delivery of RUNX1 siRNA

SC: stearyl chloride; SPC: soybean phosphatidylchloline.

## Data Availability

The data generated during the current study are available with the corresponding author on reasonable request.

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
