# Peer review of "Immunonano-Lipocarrier-Mediated Liver Sinusoidal Endothelial Cell-Specific RUNX1 Inhibition Impedes Immune Cell Infiltration and Hepatic Inflammation in Murine Model of NASH"

_ijms, 2021, doi:10.3390/ijms22168489_

Round 1

Reviewer 1 Report

Tripathi and colleagues describe the effect of a siRNA for RUNX1 using an addressable approach with vegfr3 mAb  to LSECin the reduction of inflammation and adhesion molecules expression in a murine model of induced-NASH. The hypothesis is plausible and the manuscript show a very specific and local effect that can be of interest.

However, although the effect is focused at local LSEC, NASH is thought to be an immune-mediated disease with some systemic inflammatory component that relates it with other inflammatory diseases. Have the authors check for any sign of systemic inflammation. Some discussion should be included in the discussion section.

Authors should compare their model with other obese models in mice that have been postulated for studying NAFLD/NASH pathogenesis.

Besides, there is a number of minor points to be clarified.

  1. The significant higher inflammatory grade stated in lines 72-73, page 2 is not indicated how is determined in Methods. Is there any graduation to confirm it is numeric grade and is quantified? If so, it should be explained.
  2. Subheadings 2.2, 2.3... appear to be broken with different styles.
  3. Is there any reference to the MCD model and validation as NASH model? If so, it should be indicated and referenced. If not, they should discuss, why the length of 6 months for this diet. Have they done previous validation studies and follow-up studies to set the time of the experiment at 6 months.
  4. Figure 4e. They do not indicate in the figure or in the figure legend the type of cells they are showing. In the text, authors indicate they are LSEC (line 200)
  5. Figure 5a and 5b should be quoted in text together (5a is an example and 5b the graph with numbers (line 212). Besides, the example for CD31- cells is not very clear. In the flow cytometry, it seems to be higher nomber of CD31-RUNX1+ cells in the siRNA treated mice.
  6. Materials in subheading 4.1: They do not indicate city and country for ThermoFisher and Elabsciences.
  7. Abbreviations: HSC should be indicated as hepatic stellar cells.

Reviewer 2 Report

The work of Tripathi et al. evaluated the role of RUNX1 inhibition in liver sinusoidal endothelial cells during the development of MCD-induced NASH. In their study they used a siRNA silencing approach using a vegfr3 antibody tagged immunonano-lipocarrier technique. LSEC-specific silencing of RUNX1 lead to an amelioration of typical features of NASH progression.

 Overall, the manuscript is well structured and shows very interesting new insights how LSECs can be directly targeted and how RUNX1 interactions positively influence NAFLD development via siRNA.

However, some major points of criticism arose during reading.

  1. MCD is not used as a typical NASH model due to the tremendous weight loss animals show and which the authors described in their manuscript. The authors should discuss the fact that it is rather used as an inflammatory model in this case. In addition, a weight loss greater than 20% total body weight is in some states a stop criterion to terminate the animal experiment and should therefore be very critically assessed. Do the authors furthermore think about using another steatosis-inducing model better reflecting typical features of the metabolic syndrome?
  2. Figure 1b: it remains unclear which image shows which group. It is further surprising that, if MCD fed animals are shown in the figure, absolutely no steatosis development can be observed. This is rather untypical after for a feeding period of 6 weeks. Doubts would arise that there are issues with the model. Figure 1e shows some steatosis although the image is of very poor quality.
  3. In their Methods and Supplementary Figures, the authors mention that 12 animals were used during most of the experiments. Why are only the results of 4-5 animals shown throughout the manuscript?
  4. The authors should show the results for all feeding groups including control and MCD diet for all analysis.
  5. The displayed flow experiments need to be reanalyzed thoroughly. The authors show results for intracellular markers without mentioning an intracellular staining approach in their Methods. It is further not mentioned how the unconjugated antibody was secondary labeled to be analyzed via flow cytometry. The full gating strategy should be shown. Did the authors work with an FMO to decide where they set the gates? The gating looks rather random in Figure 5 and Figure 6. Figure 5b shows a percentage of cells but it remains elusive what the percentage refers to. In Figure 5c the MFI values should be displayed as actual values. Showing a fold induction is rather untypical.
  6. It remains unclear what concentration and the carrier substance of the i.v. injections are. They just state that 3 injections end up in a concentration of 12ug/animal.

Minor points:

  1. Abbreviations such as NPC (line 80) have to be introduced when used for the first time.

Reviewer 3 Report

In their manuscript entitled “Immunonano-lipocarrier mediated Liver Sinusoidal Endothelial Cell-specific RUNX1 inhibition impedes immune cell infiltration and hepatic inflammation in murine model of NASH”, Tripathi et al. develop an immunonano-lipocarrier to specifically deplete RUNX1 in LSEC. Since RUNX1 is upregulated in livers of NASH mouse models as well as in patients, its depletion might improve NASH pathophysiology. The authors use the MCD diet model to show preliminary data on the efficacy of RUNX1-siRNA Immunonano-lipocarrier.

While the concept of the manuscript might be interesting, the validation of the lipocarrier’s specificity is not clear compromising the authors’ conclusions.  

Major concerns:

  1. In Figure 1 the authors show that the expression of RUNX1 is specifically elevated in LSEC and not HSCs. The isolation of these liver cell types is very tricky and not straightforward at all. The authors should provide more evidence of the purity of their LSEC and HSCs preps. Unfortunately, the methods section does not shed any light on this very important point. Otherwise, the conclusion that RUNX1 is specifically elevated in LSEC is questionable. Is this conclusion supported by other studies as well?
  2. Along the same lines, the authors show co-localization of RUNX1 and Vegfr3 in figure 1f. however, among dozens of Vegfr3 positive cells, only 3 are co-stained with RUNX1.
  3. In figure 3, the authors do not provide any evidence on the expression levels of RUNX1 following treatment of hepatocytes (or other liver cell types) with RUNX1-siRNA NLC.
  4. Figure 4b is used to show that the fluorescent NLCs are localized in the sinusoid. However, it is not clear at all from the images that there is indeed co-localization of the fluorescent dye and Vegfr2.
  5. The in-vivo data is not sufficient to convince that RUNX1-siRNA NLC treatment ameliorates NASH phenotype.

Round 2

Reviewer 1 Report

I consider that authors have addressed satisfactorily the questions raised.

Author Response

We are thankful to the reviewers for giving their valuable comments and helping us to improve the manuscript

Reviewer 2 Report

The authors clearly improved the manuscript in its revised version and addressed all raised concerns appropriately. I suggest to accept the manuscript in its revised version.

Author Response

The revised manuscript has been checked for grammatical errors.

We are thankful to the reviewers for giving their valuable comments and helping us to improve the manuscript.

Reviewer 3 Report

  1. Please provide more information on how the RNA was isolated from the LSEC/HSC. Were the cells kept in culture before RNA isolation, or was the RNA immediately isolated after isolation? it is not clear.
  2. It would be helpful to show some expression data of LSEC/HSC marker genes to further support the purity claim. 
